

# Comparison of four calving laws to model Greenland outlet glaciers

Youngmin Choi[1], Mathieu Morlighem[1], Michael Wood[1], and Johannes H. Bondzio[1]

[1]University of California, Irvine, Department of Earth System Science, 3218 Croul Hall, Irvine, CA 92697-3100, USA

*Correspondence to:* Youngmin Choi (youngmc3@uci.edu)

**Abstract.** Calving is an important mechanism that controls the dynamics of marine terminating glaciers of Greenland. Iceberg calving at the terminus affects the entire stress regime of outlet glaciers, which may lead to further retreat and ice flow acceleration. It is therefore critical to accurately parameterize calving in ice sheet models in order to improve the projections of ice sheet change over the coming decades and reduce the uncertainty in their contribution to sea level rise. Several calving
laws have been proposed, but most of them have been applied only to a specific region and have not been tested on other glaciers, while some others have only been implemented in one-dimensional flowline or vertical flowband models. Here, we test and compare several calving laws recently proposed in the literature using the Ice Sheet System Model (ISSM). We test these calving laws on nine tidewater glaciers of Greenland. We compare the modeled ice front evolution to the observed retreat from Landsat data collected over the past 10 years, and assess which calving law has better predictive abilities for each glacier.
Overall, the von Mises tensile stress calving law is more satisfactory than other laws for simulating observed ice front retreat, but new parameterizations that capture better the different modes of calving should be developed. Although the final positions of ice fronts are different for forecast simulations with different calving laws, our results confirm that ice front retreat highly depends on bed topography irrespective of the calving law employed. This study also confirms that calving dynamics needs to be plan-view or 3D in ice sheet models to account for complex bed topography and narrow fjords along the coast of Greenland.

## 1 Introduction

Mass loss from marine terminating glaciers along coastal Greenland is a significant contributor to global sea-level rise. Calving is one of the important processes that control the dynamics, and therefore the discharge, of these glaciers (e.g., Cuffey and Paterson, 2010; Rignot et al., 2013; Bondzio et al., 2017). Ice front retreat by enhanced calving reduces basal and lateral resistive stresses, resulting in upstream thinning and acceleration, which may lead to a strong positive feedback on glacier
dynamics (e.g. Gagliardini et al., 2010; Choi et al., 2017). Recent observations have shown that many outlet glaciers along the coast of Greenland are currently experiencing significant ice front retreat (e.g., Howat et al., 2008; Moon and Joughin, 2008). It is therefore important to accurately parameterize calving in ice sheet models in order to capture these changes and their effect on upstream flow and, consequently, improve the projections for future global sea level.

    The first attempts to model calving dynamics focused on empirical relationships between frontal ablation rate and external
variables such as water depth (Brown et al., 1982) or terminus height (Pfeffer et al., 1997). Later studies (van der Veen, 2002; Vieli et al., 2001, 2002) included ice properties and dynamics to specify calving front position. In these studies, the ice front



position is based on a height-above-buoyancy criterion (HAB), with which numerical models were able to reproduce more complex observed behaviors of Arctic glaciers. This criterion, however, was not suitable for glaciers with floating ice shelves and failed to reproduce seasonal cycles in ice front migration (Benn et al., 2007; Nick et al., 2010). Benn et al. (2007) introduced a crevasse-depth criterion, which defines calving front position where the surface crevasses reach the waterline. Nick et al.

(2010) modified this crevasse-depth criterion (CD) by including basal crevasses and their propagation for determining calving front position in a flowline model. This model successfully reproduced observed changes of several glaciers (Otero et al., 2010; Nick et al., 2012) and simulated future changes of main outlet glaciers of Greenland (Nick et al., 2013). Levermann et al. (2012) proposed to define the calving rate as proportional to the product of along and across-flow strain rates (eigencalving; EC) for Antarctic glaciers. This calving law showed encouraging results for some large Antarctic ice shelves, such as Larsen,

Ronne and Ross, but this parameterization has not been applied to Greenland glaciers, which terminate in long and narrow fjords. Morlighem et al. (2016) proposed a calving parameterization based on von Mises tensile stress (VM) to model Store glacier, Greenland. This law only relies on tensile stresses and does not include all of the processes that may yield to calving (such as damage, hydro-fracture, or bending), but it has shown encouraging results on some Greenland glaciers (Morlighem et al., 2016; Choi et al., 2017). Recently, several studies have developed new approaches based on a continuum damage model

(Duddu et al., 2013; Albrecht and Levermann, 2014) or linear elastic fracture mechanics (LEFM) (Yu et al., 2017), and Krug et al. (2014) combined damage and fracture mechanics to model calving dynamics in Greenland. These studies investigated fracture formation and propagation involved in calving, but have only focused so far on individual calving events in small-scale cases, and it is not clear how to extend these studies to three-dimensional large scale models of Greenland.

While all of these parameterizations have been tested on idealized or single, real-world geometries, most of them have not

yet been tested on a wide range of glaciers and some of these laws have only been implemented in one-dimensional flowline or vertical flowband models (Vieli and Nick, 2011). The main objective of this study is to test and compare some of these calving laws on nine different Greenland outlet glaciers using a 2D plan-view ice sheet model. Modeling ice front dynamics in a 2D horizontal or 3D model has been shown to be crucial, as the complex three-dimensional shape of the bed topography exerts an important control on the pattern of ice front retreat, which cannot be parameterized in flowline or flowband models

(e.g. Morlighem et al., 2016; Choi et al., 2017). We do not include continuum damage models and the LEFM approach in this study because these laws require to model individual calving events, whereas we focus here on laws that provide an "average" calving rate, or a calving front position, without the need to track individual calving events. While these approaches remain extremely useful to derive new parameterizations, their implementation in large scale models is not yet possible due to the level of mesh refinement required to track individual fractures.

We implement and test four different calving laws, namely the height-above-buoyancy criterion (HAB, Vieli et al., 2001), the crevasse-depth calving law (CD, Otero et al., 2010; Benn et al., 2017), the eigencalving law (EC, Levermann et al., 2012) and von Mises tensile stress calving law (VM, Morlighem et al., 2016), and model calving front migration of nine tidewater glaciers of Greenland for which we have a good description of the bed topography (Morlighem et al., 2017). The glaciers of this study are three branches of Upernavik Isstrøm (UI), Helheim glacier, three sectors of Hayes glacier, Kjer, and Sverdrup glaciers (Fig.

1). Each of these four calving laws includes a calibration parameter that is manually tuned for each glacier. These parameters



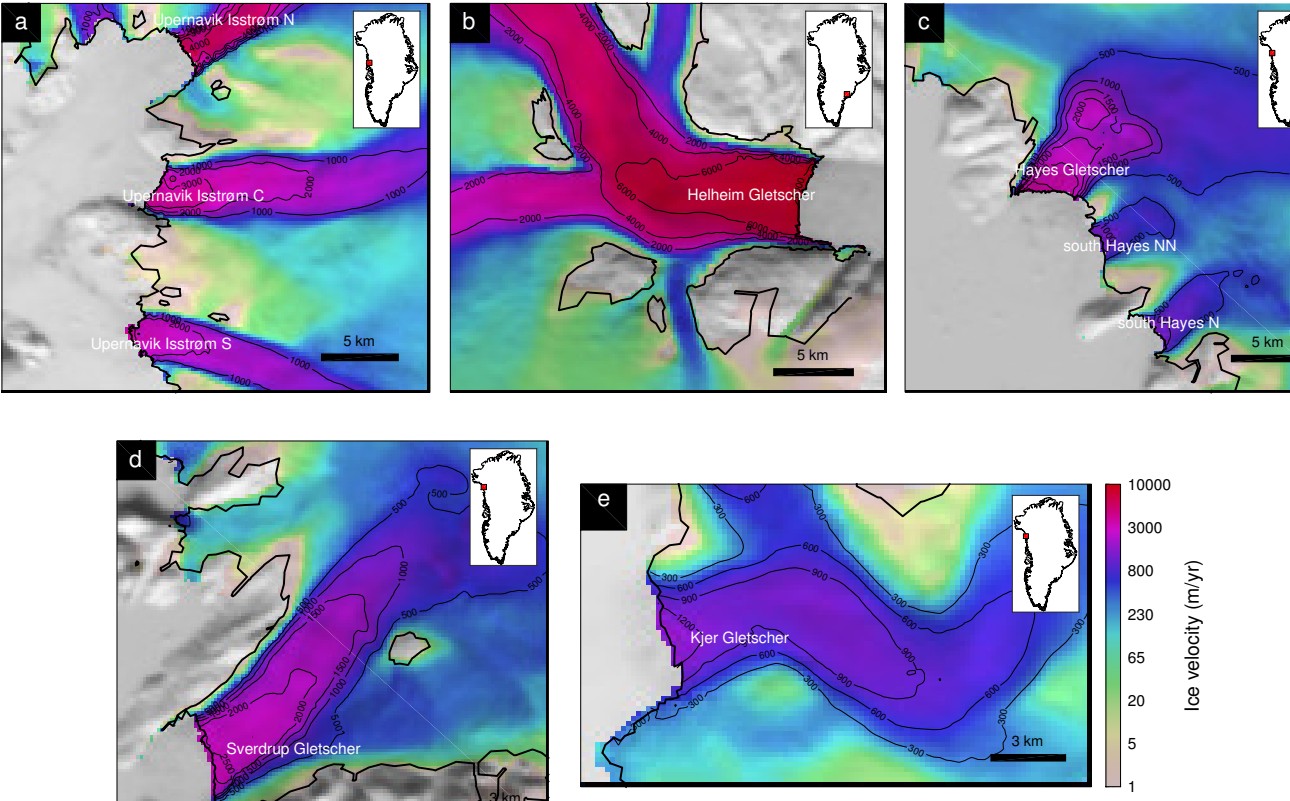

**Figure 1.** Ice surface velocity (black contours) for study glaciers (a) Upernavik Isstrøm (b) Hayes (c) Helheim (d) Sverdrup (e) Kjer. The thick black line is the ice edge.

are assumed to be constant for each glacier. To calibrate this parameter, we first model the past 10 years (2007-2017) using each calving law and compare the modeled retreat distance to the observed retreat distance. Once a best set of parameters is found, we run the model forward with the current ocean and atmospheric forcings held constant to investigate the impact of the calving laws on forecast simulations. We discuss the differences between results obtained with different calving laws for

5  the hindcast and forecast simulations and the implications thereof for the application of the calving laws to real glacier cases.

## 2   Data and method

We use the Ice Sheet System Model (ISSM, Larour et al., 2012) to implement four calving laws and to model nine glaciers. Our model relies on a Shelfy-Stream Approximation (Morland and Zainuddin, 1987; MacAyeal, 1989), which is suitable for fast outlet glaciers of Greenland (Larour et al., 2012). We use the surface elevation and bed topography data from BedMachine

10 Greenland version 3 (Morlighem et al., 2017). The nominal date of this dataset is 2008, which is close to our starting time of 2007. The surface mass balance (SMB) is from the regional atmospheric model RACMO2.3 (Noël et al., 2015) and is kept

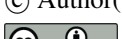



constant during our simulations. We invert for the basal friction to initialize the model, using ice surface velocity derived from satellite observations acquired at a similar period (2008-2009) (Rignot and Mouginot, 2012).

ISSM relies on the level set method (Bondzio et al., 2016) to track the calving front position. We define a level set function, $\varphi$, as being positive where there is no ice (inactive) and negative where there is ice (active region) and the calving front is

implicitly defined as the zero contour of $\varphi$. Here, we implement two types of calving laws: EC and VM provide a calving rate, $c$, whereas HAB and CD provide a criterion that defines where the ice front is located. These two types of law are implemented differently within the level set framework of ISSM.

When a calving rate is provided, the level set is is advected following the velocity of the ice front ($\mathbf{v}_{\text{front}}$) defined as a function of the ice velocity vector, $\mathbf{v}$, calving rate, $c$, and the melting rate at the calving front, $\dot{M}$:

$$\mathbf{v}_{\text{front}} = \mathbf{v} - \left(c + \dot{M}\right)\mathbf{n} \tag{1}$$

where $\mathbf{n}$ is a unit normal vector that points outward from the ice.

EC defines $c$ as proportional to strain rate along ($\epsilon_{\parallel}$) and transversal ($\epsilon_{\perp}$) to horizontal flow (Levermann et al., 2012):

$$c = K \cdot \epsilon_{\parallel} \cdot \dot{\epsilon}_{\perp} \tag{2}$$

where $K$ is a proportionality constant that captures the material properties relevant for calving. $K$ is the calibration parameter

of this calving law.

In VM, $c$ is assumed to be proportional to the tensile von Mises stress, $\tilde{\sigma}$, which only accounts for the tensile component of the stress in the horizontal plane:

$$c = \|\mathbf{v}\| \frac{\tilde{\sigma}}{\sigma_{\text{max}}} \tag{3}$$

with

$$\tilde{\sigma} = \sqrt{3}\, B\, \tilde{\dot{\varepsilon}}_e^{1/n} \tag{4}$$

where $\sigma_{\text{max}}$ is a stress threshold that is calibrated, $B$ is the ice viscosity parameter, $n = 3$ is Glen's exponent, and $\tilde{\dot{\varepsilon}}_e$ is the effective tensile strain rate defined as:

$$\tilde{\dot{\varepsilon}}_e^2 = \frac{1}{2}\left(\max\left(0, \dot{\varepsilon}_1\right)^2 + \max\left(0, \dot{\varepsilon}_2\right)^2\right) \tag{5}$$

where $\dot{\varepsilon}_1$ and $\dot{\varepsilon}_2$ are the two Eigenvalues of the 2D horizontal strain rate tensor (Morlighem et al., 2016).

For HAB and CD, we proceed in two steps at each time iteration as they do not provide explicit calving rates, $c$. First, the ice front is advected following Eq. (1) assuming that $c = 0$ and $\dot{M} \neq 0$, which simulates an advance or a retreat of the calving





front without any calving event. The calving front position is then determined by examining where the condition of each law is met. The level set, $\varphi$, is explicitly set to +1 (no ice) or -1 (ice) on each vertex of our finite element mesh depending on that condition.

For HAB, the ice front thickness in excess of floatation cannot be less than the fixed height-above-buoyancy threshold, $H_O$

(Vieli et al., 2001):

$$H_O = (1+q)\frac{\rho_w}{\rho_i}D_w \tag{6}$$

where $\rho_w$ and $\rho_i$ are the densities of sea water and ice, respectively, and $D_w$ is the water depth at the ice front, here represented by the bed depth below sea level. The fraction $q \in [0,1]$ of the floatation thickness at the terminus is our calibration parameter.

For CD, the calving front is defined as where the surface crevasses reach the waterline or surface and basal crevasses join

through the full glacier thickness. The depth of surface ($d_s$) or basal ($d_b$) crevasses is estimated from the force balance between tensile stress in the along-flow direction or any direction, water pressure in the crevasse and the lithostatic pressure:

$$d_s = \frac{R}{\rho_i g} + \frac{\rho_w}{\rho_i}d_w \tag{7}$$

$$d_b = \frac{\rho_i}{\rho_p - \rho_i}\left(\frac{R}{\rho_i g} - H_{\text{ab}}\right) \tag{8}$$

where $R$ is the resistive stress, $g$ is the gravitational acceleration, $H_{\text{ab}}$ is the height above floatation and $d_w$ is the water height

in the crevasse, which allows the crevasse to penetrate deeper (van der Veen, 1998). The water depth in the crevasse ($d_w$) is the calibration parameter of this calving law. In this study, we use two different estimations for the resistive stress, $R$. First, we use the stress only in the ice-flow direction to estimate $R$ in which changes in direction are taken into account (Otero et al., 2010). The other estimation for $R$ is the largest principal component of deviatoric stress tensor to account for tensile stress in any direction (Todd et al., 2018; Benn et al., 2017). We here use the term 'CD1' (flow direction) and 'CD2' (all directions),

respectively, to refer to these two estimations for $R$.

We use the frontal melt parameterization from Rignot et al. (2016) to estimate $\dot{M}$ in Eq. (1). The frontal melt rate, $\dot{M}$, depends on subglacial discharge, $q_{sg}$ and ocean thermal forcing, TF, defined as the difference in temperature between the potential temperature of ocean and the freezing point of seawater, as:

$$\dot{M} = \left(A h q_{sq}^{\alpha} + B\right)\text{TF}^{\beta} \tag{9}$$

where $h$ is the water depth, $A = 3 \times 10^{-4}$ m$^{-\alpha}$ day$^{\alpha-1}$ °C$^{-\beta}$, $\alpha$ = 0.39, $B$ = 0.15 °C$^{-\beta}$, and $\beta$ = 1.18. We use ocean temperature from the Estimating the Circulation and Climate of the Ocean, Phase 2 (ECCO2) project (Rignot et al., 2012). To estimate subglacial discharge, we integrate the RACMO2.3 runoff field over the drainage basin assuming that surface runoff is the dominant source of subglacial fresh water in summer (Rignot et al., 2016).



**Table 1.** Chosen calibration parameters. The values in brackets are the range of calibration parameters that produce a qualitatively similar ice front retreat pattern as the chosen calibration parameter

| Glaciers | Calving calibration parameter | | | | |
|---|---|---|---|---|---|
| | $q$ of HAB $\times 10^{-2}$(unitless) | $K$ of EC $\times 10^{-11}$ (m × a) | $d_w$ of CD1 (m) | $d_w$ of CD2 (m) | $\sigma_{\max}$ of VM (kPa) |
| Upernavik N | 5.5 | 82 | 61 | 53 | 825 |
| Upernavik C | 0.6 | 1700 | 47 | 25 | 1400 [1100 1800] |
| Upernavik S | 4 [3 4] | 8 [6.5 8.9] | 36 [35 36] | 25 | 600 [590 670] |
| Hayes | 9.1 | 35 | 45 | 47 [43 47] | 500 |
| Hayes NN | 0 | 400 [160 940] | 30 [30 31] | 23 | 1000 [0 3000] |
| Hayes N | 5.8 [4.5 5.9] | 1200 [730 2050] | 30 [20 40] | 20 [18 30] | 1000 [430 3000] |
| Helheim | 3.2 | 103 | 60 | 45 | 900 [890 910] |
| Sverdrup | 35.6 | 10 | 44 | 40 | 510 |
| Kjer | 6.3 [4.8 6.5] | 720 | 39 [38 39] | 27 | 2900 [2660 3000] |

We determine each calibration parameter (Table 1) by simulating the ice front change between 2007 and 2017 and compare the modeled pattern of retreat to observed retreat. We manually adjust these parameters for each calving law and for each basin to qualitatively best capture the observed variations in ice front position. In order to compare modeled ice front dynamics with observations, we estimate the retreat distance along five flowlines across the calving front of each glacier so that we are able to

5 account for potential asymmetric ice front retreats. Based on our calibrated models, we run the models forward until 2100 to investigate and compare the influence of different calving parameterizations on future ice front changes. For better comparison, we keep other factors (e.g., SMB, basal friction) constant in our runs. We also keep our ocean thermal forcing (eq. 9) the same as the last year of the hindcast simulation (2016-2017) until the end of our forecast simulations. The simulations are therefore divided into two time intervals: the hindcast period (2007-2017) that we use to calibrate the tuning parameters of the different

calving laws, and the forecast time period (2018-2100).

## 3 Results

The observed and modeled ice front evolutions in our simulations are shown in Figs. 2-6. The modeled retreat distances along five flowlines are compared to observed retreat distances in Fig. 7. We first notice that, in all cases, the calving laws that model a calving rate (EC and VM) have a smoother calving front than other laws. This results from the numerical implementation of

15 these laws in which it is only required to solve the advection equation of the calving front, and does not rely on a local post processing step that may yield to a more irregular shape of the calving front.



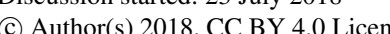


**Figure 2.** (a) The observed ice front positions between 2007-2017 and (b)-(f) modeled ice front positions obtained with different calving laws between 2007-2100 overlaid on the bed topography of Upernavik Isstrøm. The white lines are the flowlines used to calculate retreat or advance distance of ice front.

If we look at individual glaciers, Fig. 2a shows the observed pattern of retreat between 2007 and 2017 for the three branches of UI. The northern and southern branches have been rather stable over the past 10 years, but the central branch has retreated by 2.6 to 4 km. Figure 2b shows the pattern of modeled ice front position between 2007-2017 (hot colors) and 2017-2100 (cold colors) using HAB. We observe that the ice front in the central branch jumps upstream by about 2-3.5 km at the beginning of

5   the simulation and slows down as the bed elevation increases. The ice front starts retreating again after 2017 and stops when it reaches higher ground about 5 km upstream. The modeled northern and southern branches are stable until 2017 and the northern branch retreats significantly to another ridge upstream between 2017-2100. The modeled ice front using EC does not match the observed pattern of ice front retreat well (Fig. 2c). This approach causes the calving front to be either remarkably stable or creates an ice front with a strongly irregular shape. Figure 2d and 2e present the modeled ice front evolution using the

10   CD1 and CD2, respectively. Both models have similar ice front retreat patterns between 2007-2017, and they both overestimate



**Figure 3.** Same as Fig.2 but for Hayes Glaciers.

the retreat of the central branch compared to observations (Fig. 7). In the forecast simulations, the central branch retreats more when only the flow-direction stress is considered (Fig. 2d). However, in both cases, the ice front stops retreating at the same location on a pronounced ridge. The model that relies on VM shows a gradual terminus retreat and stabilizes at the end of 2017 (Fig. 2f). After 2017, the retreat behavior is similar to the one with the height-above-buoyancy law. We observe that HAB and VM reproduce the observed changes reasonably well, although they do not capture the exact timing of the 2007-2017 retreat (Fig. 2b and 2f).

The second region of interest is Hayes glacier. Currently, the three branches of this system rest on a topographic ridge, $\sim 300$ m below sea level, which is likely responsible for the observed stability in the position of the ice front over the past 10 years (Fig. 3a). The ice front of the northern glacier, however, has been retreating by up to 3 km from 2007 to 2014 and readvanced in 2016 and 2017. In this region, HAB produces a stable ice front for the northern (Hayes) and the southern sector (Hayes N) but the central sector (Hayes NN) retreats more than the observations by 0.5-0.7 km (Fig. 3b). After 2017, Hayes NN and Hayes N retreat only by a few km and stabilize there until the end of the simulation. The model using EC shows very little



**Figure 4.** Same as Fig.2 but for Helheim glacier.

change between 2007 and 2100 (Fig. 3c). As in the previous region, both the CD1 and CD2 show very similar results (Fig 3d and 3e). In the hindcast simulation (2007-2017), both models overestimate the retreat at the western part of the northern branch (Hayes). After 2017, Hayes and Hayes NN retreat quickly by 2.2-6 km into an overdeepening in the bed topography. The final positions of the ice front derived from two crevasse-depth laws are 5 km upstream of their initial position on higher

5 ridges further upstream. Figure 3f shows the modeled ice front evolution using VM. This model reproduces the stable ice front positions for two sectors (Hayes and Hayes N) but tends to overestimate the retreat for Hayes NN. Although, for the forecast simulation, VM results in more retreat than obtained with other laws for Hayes, the ice front ends up resting on the same ridges as the ones based on the crevasse-depth laws.

Figure 4a shows the observed ice front pattern for Helheim glacier. Since 2007, this glacier has shown a stable ice front

10 evolution, retreating or advancing only by a few km over the past 10 years (Cook et al., 2014). All calving parameterizations, except for EC, result in a stable or a little advanced ice front pattern (Fig. 4b-f), and only the VM model reproduces reasonably well the observed retreat distance from 2007 to 2017 for this region (Fig. 7g), although it never readvances. The other calving





**Figure 5.** Same as Fig.2 but for Sverdrup glacier.

laws do not capture the observed retreat distance or the shape of ice front properly with our ocean parameterization. In the forecast simulations, all model results show an advance or stable pattern of ice front evolutions at the end of 2100. The model with EC results in a significantly different shape of ice front compared to other models (Fig. 4c).

From 2007 to 2014, the mean terminus position of Sverdrup glacier (Fig. 5a) has been around a small ridge $\sim$ 300 m high.
5 In 2014, the glacier was dislodged from its sill and the glacier started to retreat. The models with HAB and EC show that the ice front jumps to the similar location to 2017 observed ice front (Fig. 5b and (c)). The glacier does not retreat much after 2017 in these two models. The two CDs tend to produce more retreat than other parameterizations after the ice front is dislodged from the ridge (Fig. 5d and 5e). The ice front retreat, after 2017, starts slowing down near another ridge 9 km upstream and the glacier stabilizes there until 2100. Only VM captures the timing of the retreat reasonably right (Fig. 5f). After 2017, the
10 forecast simulation shows that ice front retreats up to 4.5 km before slowing down at the second ridge upstream. The ice front then retreats past this ridge quickly and keeps retreating until it reaches a bed above sea level further upstream, where the retreat stops.



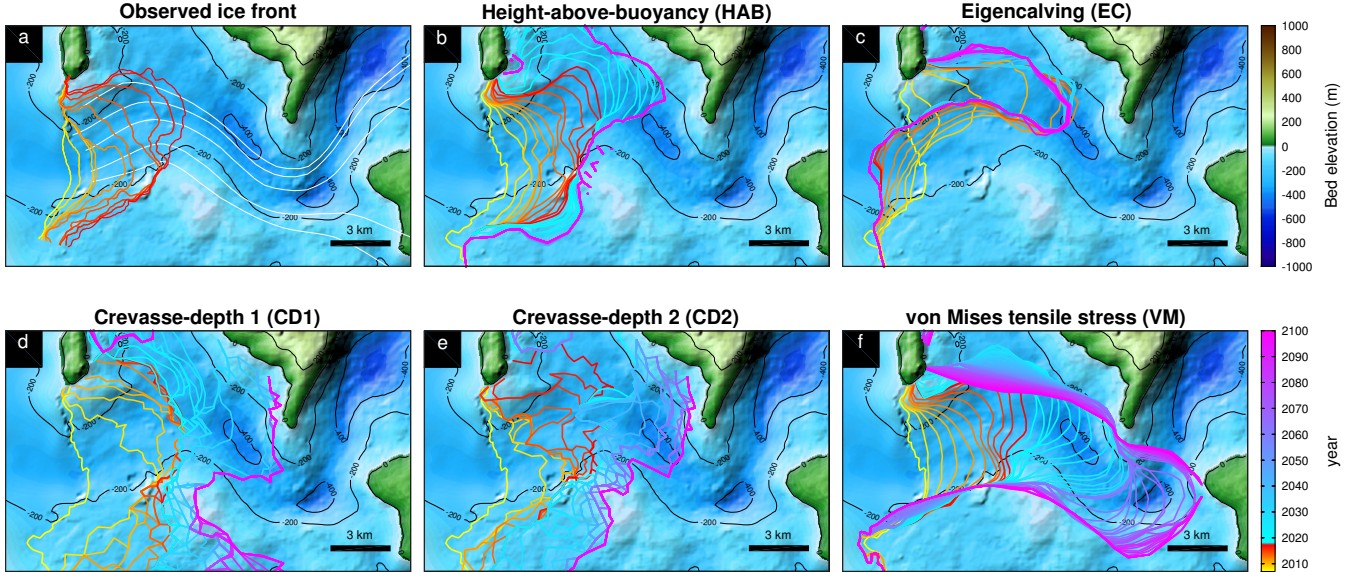

**Figure 6.** Same as Fig.2 but for Kjer glacier.

The ice front of Kjer glacier has been retreating continuously between 2007-2017 (Fig. 6a). All calving parameterizations, except for EC, simulate the observed retreat well (Fig. 6b-f and Fig. 7i). The forecast simulations, however, show different retreat patterns. HAB shows relatively less retreat than other models (Fig. 6b). The calving front slows down and stabilizes at the location where the direction of trough changes. The calving front from two crevasse-depth parameterizations retreats past

this pinning point and stops retreating at the next pinning point where the small ridge is located (Fig. 6d and 6e). In the model with VM, the retreat rate slows down near this ridge as well. The ice front, however, keeps retreating beyond this ridge and stabilizes on another ridge further upstream (Fig. 6f).

## 4   Discussion

Our results show that different calving laws produce different patterns of ice front retreat in both timing and magnitude, despite
equal climatic forcing. In the hindcast simulations, we calculate the modeled retreat distance from 2007 to 2017 for a total of 45 flowlines from our study glaciers to investigate which calving law, with the best tuning parameter, better captures the observed ice front changes (Fig. 7). We find that overall, VM captures the observed retreat better than other calving laws. For 67% (30 out of 45) of these flowlines, VM reproduce the retreat distance within 500 m from the observations, which we assume to be a reasonable range based on the seasonal variability of ice fronts, error in observations, and model resolution (Howat et al.,
2010; Bevan et al., 2012). With HAB, the modeled retreat distance is within 500 m of the observed retreat distance for 53% of the flowlines, while CD1 and CD2 capture the retreat for 51% and 40% of the the flowlines. EC reproduces only 31% of the retreat that falls into the 500-m range.



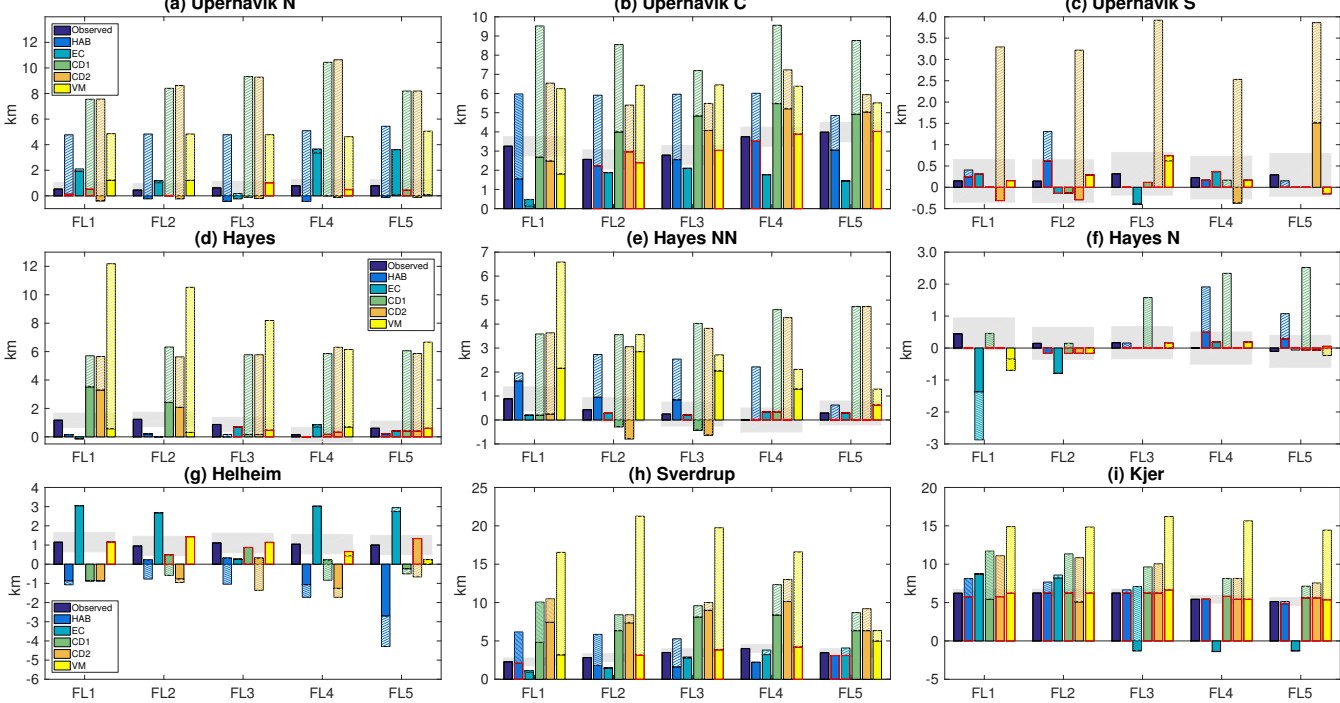

**Figure 7.** Modeled retreat distances (with respect to the calving front initial position in 2007) for different calving laws compared to observed retreat distance for nine study glaciers. The retreat distances between 2007-2017 from each calving law are shown as bar solid colors. The hatched bars are the retreated distances in 2100 for each calving law. Shaded areas represent the range of 500 m from the 2017 observed retreat and the modeled retreats that fall into this range are shown with the red edge.

EC was designed to model calving of large-scale floating shelves by including strain rates along and across ice flow (Levermann et al., 2012). Our results show that it does not work well in the case of Greenland fjords, because these glaciers flow along narrow and almost parallel valleys. The transversal strain rate, $\epsilon_\perp$, is small and noisy in these valleys, leading to a significantly different pattern of ice front changes with either a remarkably stable (e.g. Fig. 3c) or some complex shape of the modeled ice

5   front (e.g. Fig. 2c, 4c). The forecast simulations with this calving law also show different retreat patterns compared to other calving laws. While this calving law may be appropriate in the case of unconfined ice shelves, we do not recommend using this calving law for Greenland glaciers.

The two crevasse-depth calving laws are very similar in terms of the ice front retreat patterns they produce. For the regions of fast flow, the maximum principal strain rate is almost the same as the along-flow strain rate, which leads to a similar amount

10  of stress for opening crevasses. We note that for almost all of the glaciers that match the observed retreat, the model is very sensitive to the water depth in crevasses, the calibration parameter, for both laws (Table 1). Even a one meter increase in water depth significantly changes the calving rate, and thus the entire glacier dynamics. This behavior has been noticed in other modeling studies (Otero et al., 2017; Cook et al., 2012). Only one glacier (Hayes N) allows to change the water depth by up





to $\sim 18$ m and still reproduces observed ice front pattern. One reason why CDs do not capture the rate of retreat well in the hindcast simulations might also be this high sensitivity to water depth in crevasses. Models relying on this law should be taken with caution because it is hard to constrain the water depth in crevasses. The water depth in crevasses is certainly different from one year to another, and can be significantly affected by changes in surface melting and hydrology of glacier surface for the forecast simulations (e.g., Nick et al., 2013).

The model results with HAB indicate that this calving law reproduces the final position of observed calving front well for some glaciers, but does not capture continuous retreat patterns and the timing of retreat between 2007 and 2017. The ice front generally tends to jump to its final position. This may be due to the fact that we keep the height above floatation fraction ($q$) constant during our simulations. This constant fraction value also explains a relatively limited retreat compared to other calving laws for the forecast simulations. The sensitivity of the model to the parameter $q$ is different for every glaciers (Table 1). The glaciers with an ice front that is in shallow water (e.g., Hayes N, Kjer) are less sensitive to the choice of $q$ than the ones with deeper ice front. Because determining $q$ is empirical, this calving law becomes less reliable than other physics-based calving laws for the forecast simulations. Another disadvantage of this law is that it does not allow for the formation of a floating extension, and cannot be applied to ice shelves.

Our results for forecast simulations suggest that ice front retreat strongly depends on the bed topography. Although different calving laws do not always have the same final positions, the extent of glacier retreat shows a similar pattern: topographic ridges slow down or/and stop the retreat, and retrograde slopes accelerate the retreat, which has been shown in several studies (e.g., Morlighem et al., 2016; Choi et al., 2017). Whether the glaciers continue to retreat beyond these ridges depends on the calving law used and may also depend on the choice of tuning parameters. For the forecast simulations, it is not clear whether the tuning coefficients of the calving laws should be kept constant, as we did here. Some parameters potentially vary depending on future changes in external climate forcings or ice properties. These changes may affect the final locations where glaciers eventually stabilize. However, the bed topography still plays a crucial role in determining stable positions of ice fronts and the general pattern of retreat before the glaciers stabilize.

The results for Helheim glacier are very similar for all calving laws, and none of them captures the pattern of ice front migration perfectly. In the forecast simulations, the modeled ice front slightly advances until 2100 for all calving laws. This ice front advance is mostly caused by the ocean thermal forcing data used in the forecast simulations. The thermal forcing has been slightly decreasing after 2012 and a relatively cold water is applied to our forecast simulations, which leads to a similar advance of ice front for all calving law simulations. However, according to the bed topography of this region, this glacier might potentially retreat upstream if the ocean temperature increases, which may trigger more frequent calving events.

Ocean forcing is one of the limitations of this study: the frontal melt rate is simply parameterized. The ocean parameterization does not take into account ocean circulation within the fjords, which could cause localized melt higher or lower than the parameterization. We need to account for these ocean processes that may affect melt rate and could potentially vary the retreat rate of ice front. We also assume that calving front remains vertical and the melt is applied uniformly along the calving face (Choi et al., 2017). Future studies should include more detailed ocean physics and coupling to better calibrate our calving laws and improve results.





Based on our results, we recommend using the von Mises stress calving law (VM) for modeling centennial changes in Greenland tidewater glaciers within a 2D plan-view or 3D models. This calving law captures the observed pattern of retreat and rate of retreat better than other calving laws, and does allow for the formation of a floating extension. VM does not, however, necessarily capture specific modes of calving as it is only based on horizontal tensile stresses, which may be a reason why it does not always capture the pattern of ice front migration perfectly. Another disadvantage of this law is that it strongly depends on the stress threshold, $\sigma_{max}$, that needs to be calibrated. Some modeled glaciers (e.g., Helheim, Sverdrup, Kjer) are very sensitive to $\sigma_{max}$, in which case a $\sim 50$ kPa change significantly affects the calving dynamics of these glaciers (Table 1). As a result, the modeled ice front dynamics is dependent on this one single value that we keep constant through time and uniform in space, which adds uncertainty to model projections. It is therefore critical to further validate the stress threshold and improve this law by accounting for other modes of calving, or to develop new parameterizations. Current research based on discrete element models (e.g, Benn et al., 2017) or on damage mechanics (Duddu et al., 2013) may help the community derive these new parameterizations.

## 5   Conclusions

We test and compare four calving laws by modeling nine tidewater glaciers of Greenland with a 2D plan-view ice sheet model. We implement the height-above-buoyancy criterion, eigencalving law, crevasse-depth calving laws and von Mises stress calving parameterization in order to investigate how these different calving laws simulate observed front positions and affect forecast simulations. Our simulations show that the von Mises stress calving law reproduced observations better than other calving laws although it may not capture all the physics involved in calving events. Other calving laws do not capture the pattern or pace of observed retreat as well as the VM. In forecast simulations, the pattern of ice front retreat is somewhat similar for most calving laws, because of the strong control of the bed topography on ice front dynamics. Based on our results, we recommend using the tensile von Mises stress calving law, but new parameterizations should be derived in order to better capture and understand the complex processes involved in calving dynamics. It is not clear, however, whether these recommendations would apply to Antarctic ice shelves. These ice shelves calve large tabular icebergs that may be governed by different physics.

## 6   Code and data availability

The data used in this study are freely available at the National Snow and Ice Data Center, or upon request to the authors. ISSM is open source available at http://issm.jpl.nasa.gov.

*Competing interests.*   The authors declare that they have no conflict of interest.

*Acknowledgements.*   We would like to thank the editor, Olivier Gagliardini, for his constructive comments after the initial submission of this manuscript. This work was performed at the University of California Irvine under a contract with the National Science Foundation's ARCSS





program (#1504230), the National Aeronautics and Space Administration, Cryospheric Sciences Program (#NNX15AD55G) and the NASA
Earth and Space Science Fellowship Program (#80NSSC17K0409).



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
