# Peer review of "Comparison of four calving laws to model Greenland outlet glaciers"

_The Cryosphere, 2018_

## Referee Comment (RC1) · D. Benn (Referee) · 1 Aug 2018

This is an excellent paper. It makes an important contribution to the calving literature by taking the timely step of testing alternative calving laws against a large set of observations. It is well written and clearly structured, and proceeds logically towards solid conclusions. The discussion is balanced and thoughtful, and is rich in insight.

There is likely a range of dominant calving processes at the studied glaciers (e.g. melt-undercutting; super-buoyancy), and this is inherently problematical for simple calving laws. The authors' strategy of including a melt-rate parameterization alongside a calving law goes some way towards addressing this complexity, but it is clear that there is still some way to go in the search for a universal method for calculating frontal abla-

tion. The authors of course acknowledge this, and raise many important issues in the Discussion.

The impact and usefulness of the paper could be improved further if it were expanded slightly to clarify some fundamental issues associated with implementing and testing the calving laws. Two issues in particular would benefit from more detailed treatment: 1) the rationale behind model tuning; and 2) the methods employed to identify the 'best fit' between observations and the tuned models.

All of the laws - as implemented here - rely on tuning. As the authors explain, this places limits on their practical usefulness when applied to uncalibrated glaciers, or when projected into the future. The authors should also note that this is particularly problematic where the parameters span a wide range (orders of magnitude for HAB and EC compared with a factor of 2-3 for CD and VM [with one outlier in the latter]). A more fundamental point that should be made is that the ability of a tuned model to replicate observations does not prove that it 'works' in a meaningful way. The success of a calving law tuned on a glacier-by-glacier basis may simply be a test of its flexibility, as opposed to its actual predictive/diagnostic power.

The authors make some very interesting points regarding the tuning of HAB. The rationale behind HAB (as originally developed for Columbia Glacier) is that the glacier will calve as it approaches buoyancy. As noted by the authors, this does not allow floating tongues; however, the opposite is also true: HAB predicts that a glacier will not calve if HAB>Ho. But of course, many well-grounded glaciers do calve, for various reasons, meaning that HAB is problematical in both directions. A wide range of grounding conditions in the study glaciers - and associated calving processes - probably accounts for the very wide range of q in this study. This has major implications for modelling future conditions, if buoyancy conditions and dominant calving processes change through time.

The authors rightly flag up the problems with using crevasse water depth as a tuning
parameter in the CD models. I am now of the opinion that water depth is neither useful nor appropriate as a tuning parameter in most cases (see Benn et al., 2017, p. 701). (Ice shelf hydrofracture may be a significant exception.) I agree with the authors that results obtained by water-depth tuning of CD should be treated with caution (for example, I think that the studies of KNS by Lea et al. are deeply flawed for this reason).

However, it should be noted that water depth is not included in the CD model as implemented by Todd et al. (2018). In that study, the CD model was able to reproduce seasonal calving variability at Store Glacier without any tuning - in stark contrast to the performance of CD in the present paper. A major difference between Todd et al. (2018) and the present study is the model physics (3D full stress vs. 2D plan-view). Therefore, the authors could be more explicit that the CD model may not be the best choice for 2D plan-view models because they do not accurately capture the required stresses.

To aid comparison between the present study and Todd et al. (2018), I would like the authors to show results of CD with dw = 0 alongside the tuned results. A model that does not require any tuning has obvious advantages, so it would be particularly interesting to see how it performs in this case.

The performance of VM is impressive, and it is worthwhile delving deeper into possible reasons for this. The results show that, on a glacier-by-glacier basis, there tends to be a consistent relationship between calving rate and v(sigma_vm/sigma_max) (Eq. 3). Perhaps the strength of these relationships partially reflects including the velocity vector in the calving rate. At the least, the strong correlation between calving rate & velocity means that VM is inherently primed to produce more reasonable calving rates. The extent to which this questions the model's predictive capacity/skill is difficult to address, but this is clearly an issue that requires further investigation in future. This point should be added to the Discussion (p. 14, around line 5).

Optimization of the model parameters was done by manually finding the values that "qualitatively best capture the observed variations". The authors should provide more

information about this procedure. Figure 7 very usefully compares the modelled 2017 front positions, but what of the other characteristics of the records (e.g. timing of still-stands, advances or retreat episodes)? What criteria were used to decide on the best-fit parameters? Were some criteria weighted more than others? Were the criteria used consistently? To address these questions, more information should be added to the text around p. 6, line 3.

I also suggest the authors present a set of time-distance diagrams comparing observations and model results for each flowline (perhaps as Supplementary Material). This would then allow readers to assess the performance of each model in greater detail than is currently possible.

Minor points:

p. 2, L5: "This law only relies on tensile stresses..." add: "and frontal velocity"

p. 4 L26: Clarify what is meant by 'M = non-zero'. Does the method somehow require some melt rate, or is this simply intended to state that the appropriate melt rate is applied?

p. 5, Equation 9: B is already used for the ice viscosity parameter, so a different symbol is needed for the melt rate parameter.

---

## Referee Comment (RC2) · J. Bassis (Referee) · 9 Oct 2018

This study compares four quasi-empirical calving laws to assess their suitability in predicting terminus retreat from 9 Greenland tidewater glaciers. The authors optimize unknown parameters in the calving laws to best fit each glacier and then compare the best projected calving front position with observed calving front positions and to project mass loss associated with calving forward. The authors find that so-called von Mises calving is the best fitting calving law for most glaciers.

Although several studies comparing calving laws have been published in the literature, most previous comparisons have focused on flowline models. This study is one of the first to assess the behavior of different calving laws using two-dimensional (mapview) glacier geometry and is a promising first stab at this problem. Overall, I think the manuscript is quite promising and most of my comments are relatively minor or quaintly technical in nature. Here, I should also disclose, I have found myself reviewing several of the authors prior papers. I think the authors and editor should be cognizant of the fact that my comments likely overlap and they may want to discard or de-emphasize some comments to make sure that the same voice (mine) is not overly contributing to this conversation. My more detailed comments are included below:

The authors come to an interesting conclusion that the Von Mises calving law is the calving law that best describes observed changes and, hence, might be the best to use for future projections of Greenland outlet glaciers. This is an interesting result, but I would encourage the authors to dwell a little bit more on *why* this calving law seems to perform so well and to revisit the limitations associated with making projections based on tuned calving laws. The fact there is such a disparity in best fitting parameters is interesting because it implies there is no single parameter that can be plugged into a calving law that will yield adequate results. This in turn implies that parameters appropriate for one instance of time (or slice of time) may not remain valid in the future. This would significantly impact projections if the so-called best fitting parameters evolved over time.

There is a final interesting point, which is that the Von Mises calving law is fundamentally different from the other calving laws. Each of the other calving laws depends on local (scalar) properties of the glacier at (or at least near) the calving front. These laws are all essentially empirical, but also depend solely on coordinate system independent parameters of the system. The Von Mises calving law, in contrast, depends on the velocity at the calving front and velocity is not reference frame independent. For example, if I were to adopt a Lagrangian reference frame that moves with the glacier calving front, the velocity at the calving front would be exactly zero and, as far as I can tell, the calving rate would also vanish. This dependence of the calving rate on reference frame is something that theorists would find disturbing, but is less bothersome if we think of

the law as empirical and calibrated to work well in some defined parameter regime.

Another difference between the Von Mises calving law and the other laws is that the velocity dependence of the Von Mises calving law means that the calving rate is non-locally determined. Changes in faraway boundary conditions (or at least in behavior upstream from the calving front) could instantaneously propagate and affect calving rates. This "action-at-a distance" is also interesting and means that the Von Mises calving law is an integrator of glacier behavior in the vicinity of the calving front. Overall, I do wonder how much of the behavior of the model is due to the appearance of the velocity in the calving rate. I would like to hear the authors comment more on these model formulation differences partly because I think I can rationalize the velocity dependence of the Von Mises calving law as a linearization about steady-state. In this argument we start from a steady-state condition in which calving rate = terminus velocity and then linearize to deduce a velocity dependent calving rate. This linearization, however, does depend on linearizing about a steady-state and thus might explain some of the variability in inferred yield strengths. It would also hint that the calving law would remain appropriate for short periods of time, but could fail when applied to longer time periods. This comes back to my point about uncertainty in projections using a tuned parameterization.

Miscellaneous comments:

Page 4, line 25: I believe that HAB and CD models could also be implemented in such a way that they yield continuous rates. This can be done relatively easily for the HAB criterion by taking the advective derivative of ice thickness at the calving front and determining the rate of advance necessary to maintain a critical height above-buoyancy. I believe one could also do this for the CD model by relating the stress at the calving front to the ice thickness and water depth. This may (or may not) change some of the behavior of these models.

I think it would for beneficial if the authors could state in a few sentences the spatial

and temporal resolution studies they have done to make sure that results are numerically converged. I have often found that accurately simulating advance and retreat of glaciers requires far more resolution than I would have expected. I do wonder if the blocky behavior of the HAB and CD models might be reduced with finer resolution and if any of the other behavior of the models is persistent when resolution is halved or decreased by a factor of 8.

Equations 7-8: I wonder if it would be better to write these equations in terms of deviatoric rather than resistive stresses. Resistive stresses, as defined by Van der Veen, are not the same as deviatoric stresses. Here, it is unclear if deviatoric stresses (e.g., near line 15) or resistive stresses (equations 7-8) are used. Deviatoric stresses are easy to compute using a numerical model and are directly related to the rheology. Resistive stresses have an awkward factor of two difference. Resistive stresses were a useful quantity when attempting to understand which terms in the stress balance are important, but less useful when using an ice sheet model where factor of two errors often creep into calculations.

---

## Author Comment (AC1) · 6 Nov 2018

| Comparison of four calving laws to model Greenland outlet |
|-----------------------------------------------------------|
| glaciers                                                  |
|  <li>Response to reviewers –</li>                |
|                                                           |
| Youngmin CHOI et al.                                      |
| November 5, 2018                                          |
|                                                           |

6 We thank the two reviewers for their positive and constructive comments that significantly im-7 proved the manuscript. We address their remarks below point by point.

**8 1 Reviewer #1**

This is an excellent paper. It makes an important contribution to the calving literature by taking
the timely step of testing alternative calving laws against a large set of observations. It is well
written and clearly structured, and proceeds logically towards solid conclusions. The discussion is
balanced and thoughtful, and is rich in insight.

There is likely a range of dominant calving processes at the studied glaciers (e.g. melt- undercutting; super-buoyancy), and this is inherently problematical for simple calving laws. The authors' strategy of including a melt-rate parameterization alongside a calving law goes some way towards addressing this complexity, but it is clear that there is still some way to go in the search for a universal method for calculating frontal ablation. The authors of course acknowledge this, and raise many important issues in the Discussion.

The impact and usefulness of the paper could be improved further if it were expanded slightly to clarify some fundamental issues associated with implementing and testing the calving laws. Two issues in particular would benefit from more detailed treatment: 1) the rationale behind model tuning; and 2) the methods employed to identify the best fit between observations and the tuned models. 24 Thank you. We addressed these two main issues below.

All of the laws - as implemented here - rely on tuning. As the authors explain, this places limits on 25 their practical usefulness when applied to uncalibrated glaciers, or when projected into the future. 26 The authors should also note that this is particularly problematic where the parameters span a 27 wide range (orders of magnitude for HAB and EC compared with a factor of 2-3 for CD and VM 28 [with one outlier in the latter]). A more fundamental point that should be made is that the ability 29 of a tuned model to replicate observations does not prove that it works in a meaningful way. The 30 success of a calving law tuned on a glacier-by-glacier basis may simply be a test of its flexibility, 31 as opposed to its actual predictive/diagnostic power. 32

33 We agree with the reviewer. We added a paragraph about this point in the discussion section.

The authors make some very interesting points regarding the tuning of HAB. The rationale be-34 hind HAB (as originally developed for Columbia Glacier) is that the glacier will calve as it ap-35 proaches buoyancy. As noted by the authors, this does not allow floating tongues; however, the 36 opposite is also true: HAB predicts that a glacier will not calve if HAB > Ho. But of course, many 37 well-grounded glaciers do calve, for various reasons, meaning that HAB is problematical in both 38 directions. A wide range of grounding conditions in the study glaciers - and associated calving 39 processes - probably accounts for the very wide range of q in this study. This has major impli-40 cations for modelling future conditions, if buoyancy conditions and dominant calving processes 41 change through time. 42

43 We clarified this point in the discussion.

The authors rightly flag up the problems with using crevasse water depth as a tuning parameter in the CD models. I am now of the opinion that water depth is neither useful nor appropriate as a tuning parameter in most cases (see Benn et al., 2017, p. 701). (Ice shelf hydrofracture may be a significant exception.) I agree with the authors that results obtained by water-depth tuning of CD should be treated with caution (for example, I think that the studies of KNS by Lea et al. are deeply flawed for this reason).

However, it should be noted that water depth is not included in the CD model as im- plemented by
Todd et al. (2018). In that study, the CD model was able to reproduce seasonal calving variability
at Store Glacier without any tuning - in stark contrast to the performance of CD in the present
paper. A major difference between Todd et al. (2018) and the present study is the model physics
(3D full stress vs. 2D plan-view). Therefore, the authors could be more explicit that the CD model
may not be the best choice for 2D plan-view models because they do not accurately capture the
required stresses.

57 We added this study (Todd et al.(2018)) in the discussion section and explained the difference 58 between 2D plan view and 3D models.

To aid comparison between the present study and Todd et al. (2018), I would like the authors to show results of CD with dw = 0 alongside the tuned results. A model that does not require any tuning has obvious advantages, so it would be particularly interesting to see how it performs in this case.

We performed this experiment and added the corresponding figures to the Supplementary Material
 (Fig. S1-S4).

The performance of VM is impressive, and it is worthwhile delving deeper into possible reasons for 65 this. The results show that, on a glacier-by-glacier basis, there tends to be a consistent relationship 66 between calving rate and v(sigma\_vm/sigma\_max) (Eq. 3). Perhaps the strength of these relation-67 ships partially reflects including the velocity vector in the calving rate. At the least, the strong 68 correlation between calving rate & velocity means that VM is inherently primed to produce more 69 reasonable calving rates. The extent to which this questions the models predictive capacity/skill 70 is difficult to address, but this is clearly an issue that requires further investigation in future. This 71 point should be added to the Discussion (p. 14, around line 5). 72

We agree with the reviewer's point about the relationship between calving rate and ice velocity. We
 added this point in the discussion section.

Optimization of the model parameters was done by manually finding the values that "qualitatively
best capture the observed variations". The authors should provide more information about this
procedure. Figure 7 very usefully compares the modelled 2017 front positions, but what of the
other characteristics of the records (e.g. timing of still- stands, advances or retreat episodes)?
What criteria were used to decide on the bestfit parameters? Were some criteria weighted more
than others? Were the criteria used consistently? To address these questions, more information
should be added to the text around p. 6, line 3.

We only consider the retreat distance between 2007 and 2017 and do not account for the timings
of retreat or advance for choosing calibration parameters. We clarified this in the Data and Method
section.

I also suggest the authors present a set of time-distance diagrams comparing observations and
 model results for each flowline (perhaps as Supplementary Material). This would then allow read ers to assess the performance of each model in greater detail than is currently possible.

88 Done (Fig. S5-S7).

89 Minor points:

90 p. 2, L5: "This law only relies on tensile stresses..." add: "and frontal velocity"

91 Done.

- p. 4 L26: Clarify what is meant by "M = non-zero". Does the method somehow require some melt rate, or is this simply intended to state that the appropriate melt rate is applied?
- 94 We meant 'the appropriate melt rate'. We clarified the sentense.

p. 5, Equation 9: B is already used for the ice viscosity parameter, so a different symbol is needed
for the melt rate parameter.

97 Done. We changed B in the Equation 9 into b.

**98 2 Reviewer #2**

This study compares four quasi-empirical calving laws to assess their suitability in predicting terminus retreat from 9 Greenland tidewater glaciers. The authors optimize unknown parameters in the calving laws to best fit each glacier and then compare the best projected calving front position with observed calving front positions and to project mass loss associated with calving forward. The authors find that so-called von Mises calving is the best fitting calving law for most glaciers.

Although several studies comparing calving laws have been published in the literature, most pre-104 vious comparisons have focused on flowline models. This study is one of the first to assess the 105 behavior of different calving laws using two-dimensional (map view) glacier geometry and is a 106 promising first stab at this problem. Overall, I think the manuscript is quite promising and most 107 of my comments are relatively minor or quaintly technical in nature. Here, I should also disclose, 108 I have found myself reviewing several of the authors prior papers. I think the authors and editor 109 should be cognizant of the fact that my comments likely overlap and they may want to discard or 110 de-emphasize some comments to make sure that the same voice (mine) is not overly contributing to 111 this conversation. My more detailed comments are included below: 112

The authors come to an interesting conclusion that the Von Mises calving law is the calving law that best describes observed changes and, hence, might be the best to use for future projections of Greenland outlet glaciers. This is an interesting result, but I would encourage the authors to dwell a little bit more on \*why\* this calving law seems to perform so well and to revisit the limitations associated with making projections based on tuned calving laws. The fact there is such a disparity in best fitting parameters is interesting because it implies there is no single parameter that can be plugged into a calving law that will yield adequate results. This in turn implies that parameters appropriate for one instance of time (or slice of time) may not remain valid in the future. This would significantly impact projections if the so-called best fitting parameters evolved over time.

The first reviewer has the similar comments and this is an excellent point. We think that the strong relationship between calving rate and ice velocity might produce more reasonable calving rates but whether this holds for future simulations needs further investigation. We added these points in the discussion section.

There is a final interesting point, which is that the Von Mises calving law is fundamentally different 126 from the other calving laws. Each of the other calving laws depends on local (scalar) properties of 127 the glacier at (or at least near) the calving front. These laws are all essentially empirical, but also 128 depend solely on coordinate system independent parameters of the system. The Von Mises calving 129 law, in contrast, depends on the velocity at the calving front and velocity is not reference frame 130 independent. For example, if I were to adopt a Lagrangian reference frame that moves with the 131 glacier calving front, the velocity at the calving front would be exactly zero and, as far as I can 132 tell, the calving rate would also vanish. This dependence of the calving rate on reference frame 133 is something that theorists would find disturbing, but is less bothersome if we think of the law as 134 empirical and calibrated to work well in some defined parameter regime. 135

The reviewer is right: the von Mises calving law here depends on the reference frame. We adopted a Eulerian frame in this study. Theoretically, we could apply the same law as a calving criterion rather than a speed, but removing the ice where  $\tilde{\sigma} > \sigma_{max}$ , which would then be independent of the reference frame. We tried to use this law but it failed because the tensile stress is too strong in the margins, which leads to a faster retreat along the sides of the glacier than the center (See Fig. S8), which is way we ended up formulating this law in terms of calving rate by multiplying by the ice velocity.

Another difference between the Von Mises calving law and the other laws is that the velocity depen-143 dence of the Von Mises calving law means that the calving rate is non-locally determined. Changes 144 in faraway boundary conditions (or at least in behavior upstream from the calving front) could 145 instantaneously propagate and affect calving rates. This "action-at-a distance" is also interesting 146 and means that the Von Mises calving law is an integrator of glacier behavior in the vicinity of the 147 calving front. Overall, I do wonder how much of the behavior of the model is due to the appear-148 ance of the velocity in the calving rate. I would like to hear the authors comment more on these 149 model formulation differences partly because I think I can rationalize the velocity dependence of 150 the Von Mises calving law as a linearization about steady-state. In this argument we start from 151

a steady-state condition in which calving rate = terminus velocity and then linearize to deduce a
velocity dependent calving rate. This linearization, however, does depend on linearizing about a
steady-state and thus might explain some of the variability in inferred yield strengths. It would also
hint that the calving law would remain appropriate for short periods of time, but could fail when
applied to longer time periods. This comes back to my point about uncertainty in projections using
a tuned parameterization.

We agree with the reviewer's point that changes upstream may affect calving rates if the changes 158 are large. We limit the maximum calving rate to 3 km/yr, which can prevent unrealistic calving 159 rates caused by abrupt changes upstream from the ice front that may be short lived. We added 160 this point in the Data and method section. We added the figure in the supplementary material 161 (Fig.S8) to show the effect of velocity components in the calving rate. In addition, we agree with 162 the point about uncertainty in projections using a tuned parameterization, which is now disscused 163 in the discussion section. We were not able, however, to show that the calving rate is a linearization 164 around steadystate. 165

Miscellaneous comments: Page 4, line 25: I believe that HAB and CD models could also be implemented in such a way that they yield continuous rates. This can be done relatively easily for the HAB criterion by taking the advective derivative of ice thickness at the calving front and determining the rate of advance necessary to maintain a critical height above buoyancy. I believe one could also do this for the CD model by relating the stress at the calving front to the ice thickness and water depth. This may (or may not) change some of the behavior of these models.

This is a good point and we thought about implementing HAD and CD calving laws as continuous calving rates like EC and VM calving laws. However, it proved to be significantly more complicated in 2D plan-view/3D models than in a flowline model. The parallel architecture of ISSM adds further complications and we decided to not look further into it. The idea of this paper was also to compare published calving laws and so we still think that testing the calving law as they were introduced in the literature is helpful.

I think it would for beneficial if the authors could state in a few sentences the spatial and temporal
resolution studies they have done to make sure that results are numerically converged. I have often
found that accurately simulating advance and retreat of glaciers requires far more resolution than
I would have expected. I do wonder if the blocky behavior of the HAB and CD models might be
reduced with finer resolution and if any of the other behavior of the models is persistent when
resolution is halved or decreased by a factor of 8.

The spatial and temporal resolutions are now explained in the Data and method section. We use the time steps that satisfy the Courant-Friedrichs-Lewy condition [*Courant et al.*, 1928] for each glacier to make sure that the solutions are temporally converged. We also added Fig.S9 to show that our results do not change significantly as we change the mesh resolution.

Equations 7-8: I wonder if it would be better to write these equations in terms of deviatoric rather 188 than resistive stresses. Resistive stresses, as defined by Van der Veen, are not the same as deviatoric 189 stresses. Here, it is unclear if deviatoric stresses (e.g., near line 15) or resistive stresses (equations 190 7-8) are used. Deviatoric stresses are easy to compute using a numerical model and are directly 191 related to the rheology. Resistive stresses have an awkward factor of two difference. Resistive 192 stresses were a useful quantity when attempting to understand which terms in the stress balance 193 are important, but less useful when using an ice sheet model where factor of two errors often creep 194 into calculations. 195

196 We use the deviatoric stress to calculate the along-flow and the largest stresses for two crevasse 197 depth calving laws as in *Otero et al.* [2010]. We now use the symbol  $\sigma$  instead of R in Eq. (7) and 198 (8) and clarify this in the text.

**References**

- Courant, R., K. Friedrichs, and H. Lewy, Über die partiellen differenzengleichungen der mathema tischen physik, *Mathematische Annalen*, *100*, 32–74, doi:10.1007/BF01448839, 1928.

[revised manuscript text omitted]